# Different Responses of Soil Bacterial and Fungal Communities in Three Typical Vegetations following Nitrogen Deposition in an Arid Desert

**DOI:** 10.3390/microorganisms11102471

**Published:** 2023-10-01

**Authors:** Zhihao Zhang, Gangliang Tang, Xutian Chai, Bo Liu, Xiaopeng Gao, Fanjiang Zeng, Yun Wang, Bo Zhang

**Affiliations:** 1Xinjiang Key Laboratory of Desert Plant Roots Ecology and Vegetation Restoration, Xinjiang Institute of Ecology and Geography, Chinese Academy of Sciences, Urumqi 830011, China; zhangzh@ms.xjb.ac.cn (Z.Z.);; 2State Key Laboratory of Desert and Oasis Ecology, Key Laboratory of Ecological Safety and Sustainable Development in Arid Lands, Xinjiang Institute of Ecology and Geography, Chinese Academy of Sciences, Urumqi 830011, China; 3Cele National Station of Observation and Research for Desert-Grassland Ecosystems, Cele 848300, China; 4University of Chinese Academy of Sciences, Beijing 100049, China; 5School of Resources and Environment, Linyi University, Linyi 276000, China; 6Department of Soil Science, University of Manitoba, Winnipeg, MB R3T 2N2, Canada; 7Life Science and Technology School, Linnan Normal University, Zhanjiang 524048, China; 8National Engineering Technology Research Center for Desert-Oasis Ecological Construction, Xinjiang Institute of Ecology and Geography, Chinese Academy of Sciences, Urumqi 830011, China

**Keywords:** nitrogen deposition, microbial community, plant community types, desert steppe, global change

## Abstract

The effects of increased nitrogen (N) deposition on desert ecosystems have been extensively studied from a plant community perspective. However, the response of soil microbial communities, which play a crucial role in nutrient cycling, to N inputs and plant community types remains poorly understood. In this study, we conducted a two-year N-addition experiment with five gradients (0, 10, 30, 60, and 120 kg N ha^−1^ year^−1^) to evaluate the effect of increased N deposition on soil bacterial and fungal communities in three plant community types, namely, *Alhagi sparsifolia* Shap., *Karelinia caspia* (Pall.) Less. monocultures and their mixed community in a desert steppe located on the southern edge of the Taklimakan Desert, Northwest China. Our results indicate that N deposition and plant community types exerted an independent and significant influence on the soil microbial community. Bacterial α-diversity and community dissimilarity showed a unimodal pattern with peaks at 30 and 60 kg N ha^−1^ year^−1,^ respectively. By contrast, fungal α-diversity and community dissimilarity did not vary significantly with increased N inputs. Furthermore, plant community type significantly altered microbial community dissimilarity. The Mantel test and redundancy analysis indicated that soil pH and total and inorganic N (NH_4_^+^ and NO_3_^−^) levels were the most critical factors regulating soil microbial communities. Similar to the patterns observed in taxonomic composition, fungi exhibit stronger resistance to N addition compared to bacteria in terms of their functionality. Overall, our findings suggest that the response of soil microbial communities to N deposition is domain-specific and independent of desert plant community diversity, and the bacterial community has a critical threshold under N enrichment in arid deserts.

## 1. Introduction

In recent decades, increased nitrogen (N) fertilizer use and fossil energy consumption have contributed significant amounts of reactive N to the atmosphere, of which 60% is returned to the soil [1,2]. As one of the drivers of global change, N deposition profoundly affects plants and microorganisms in various ecosystems, including deserts [3,4]. Soil microorganisms play a vital role in biogeochemical cycles, plant nutrient uptake, and climate regulation. The differential responses of plants to N deposition at the individual and community levels alter the soil microbial communities by changing litter properties and plant identity, diversity, and root exudates [5,6,7,8]. In this context, ecosystem services provided by soil microorganisms may be profoundly affected [9]. To fully understand the potential influence of global change on soil ecosystems, soil microbial communities need to be investigated under different N deposition and plant community types.

Deserts cover one-third of the global land surface and have a lower critical N load than other ecosystems [10,11,12]. N deposition can affect soil microbial diversity and function by favoring certain nitrogen-use species and suppressing others [13]. This disruption can lead to imbalances in nutrient cycling processes, increasing N leaching and reducing nutrient availability to plants [11], negatively impacting ecosystem productivity, and potentially affecting ecosystem functioning and resilience [3]. A four-year field experiment showed that N deposition significantly altered soil bacterial communities by increasing the abundance of ammonia-oxidizing bacteria and reducing denitrification genes in a desert steppe in Inner Mongolia, China [14]. A recent study conducted in the Gurbantunggut Desert, Northwest China, showed that six-year N additions significantly altered the structure of soil bacterial and fungal community composition by changing soil conditions such as pH [15]. However, in the same desert, two-year N additions slightly affected soil microbial community composition [16]. Similarly, She et al. [17] and Huang et al. [18] reported no significant effect of N deposition on the soil microbial community in the Mu Us Desert, China. These inconsistent results may be due to climate, the amount and duration of N treatment, and initial soil properties [19]. In addition, soil bacterial and fungal communities behave differently when faced with the same environmental changes. Compared to the fungal community, the bacterial community is more tolerant to N deposition in subtropical and arid regions [4,15,20]. These studies suggest that the response of the soil microbial community to N deposition is microbial-domain-specific. Currently, there is no clear understanding of how soil microorganisms respond to N deposition in typical desert plant communities at the southern edge of the Taklamakan Desert in extremely arid areas. In this area, the vegetation consists mainly of deep-rooted plants such as *Alhagi sparsifolia* Shap. and *Karelinia caspia* (Pall.) Less.

Previously, we conducted an intercropping experiment with three plant community types (or two diversity levels), namely, *A. sparsifolia* and *K. caspia* monocultures and their mixed community [21]. We showed that the microbial biomass C/N in the mixed community (14.1 ± 0.8, mean ± standard deviation) was significantly higher than that in the monoculture communities (8.9 ± 0.6 in *A. sparsifolia* monoculture and 9.3 ± 0.7 in *K. caspia* monoculture). This observation suggests that N is a critical limiting factor for the microbial community in this area. Therefore, we hypothesized that (1) increased N inputs will significantly shape the soil microbial community composition by improving microbial species diversity; and (2) this change will be partly attributed to changes in plant community characteristics caused by N deposition, i.e., N deposition interacting with plant community type exerts a significant influence on the soil microbial community. To test the above hypotheses, we used high-throughput sequencing to characterize soil bacterial and fungal communities using a two-year N-addition treatment with five gradients in a desert steppe at the southern edge of the Taklimakan Desert, Northwest China. This study aimed to provide a scientific basis for accurately predicting the response of the soil microbial community to excess N enrichment.

## 2. Materials and Methods

### 2.1. Site Description

This study was conducted in a desert steppe (80°43′ E, 37°00′ N, 1363 m asl) at the southern edge of the Taklimakan Desert, China, the second-largest drifting desert in the world (Figure 1a). The mean annual temperature, average annual precipitation and potential evaporation in this region are 12.7 °C, 35 mm and 2600 mm, respectively. Rainfall occurs from May to August. The groundwater depth in the study area ranges from 13 to 15 m. The sandy soil has a bulk density of 1.36 g/cm^3^ [22]. The vegetation is sparsely covered, ranging from 20% to 40%, and dominated by perennial plants. Among these, the most common vegetation types in this area are *A. sparsifolia* (legume) monoculture, *K. caspia* (Compositae) monoculture and their mixed communities [23,24]. *A. sparsifolia* has a height of 0.5–0.9 m, a canopy size of 0.6–1.6 m^2^, and a rooting depth of 13–15 m. On the other hand, *K. caspia* has a height of 0.5–0.9 m, a canopy size of 0.6–1.6 m^2^, and a rooting depth of 3–9 m. Since 1983, this area has been a natural desert steppe and has been fenced off.

### 2.2. Experimental Design

Urea (CON_2_H_4_) has been used to simulate N deposition since May 2019. We applied urea at the levels of 0 (CK),10 (N10), 30 (N30), 60 (N60), and 120 kg N ha^−1^ year^−1^ (N120). The natural levels of N deposition at the study site are a considerably low 6.33 kg N ha^−1^ year^−1^ [24] and were not considered in this work to explore the influence of N deposition on soil microbial community. The above five N treatments were conducted on *A. sparsifolia* monoculture, *K. caspia* monoculture and their mixed communities (Figure 1b), and three 6 m × 6 m repeating plots spaced 3 m apart were set for each treatment. The distances among the three community types are less than 200 m. A total of 45 plots were used, i.e., 3 plant community types × 5 N-addition treatments × 3 duplications. Urea dissolved in 5 L of purified water was evenly sprayed over plots with a hand-held sprayer at the beginning of each month (May to October). By contrast, an equal amount of purified water was sprayed on the control plots. According to weather station monitoring at Cele National Station, about 1 km from our study site, annual precipitations in 2019 and 2020 were 47.2 mm and 63.2 mm, respectively. Before experimental treatment, the properties of soil were as follows: soil organic carbon (SOC), 1.50 g kg^−1^; total nitrogen (TN), 0.25 g kg^−1^; nitrate (NO_3_^−^−N), 19.74 mg kg^−1^; ammonium (NH_4_^+^−N), 3.84 mg kg^−1^; available nitrogen (AN), 28.87 mg kg^−1^; available phosphorus (AP), 5.07 mg kg^−1^; available potassium (AK), 223.61 mg kg^−1^; and pH, 7.11.

### 2.3. Soil Sampling

Soil samples (0−20 cm) were collected from each plot on September 20, 2020. Before the collection of soil samples, the litter layer was removed. In each plot, three soil cores were collected along the diagonal, including under-shrub patches and the interspaces (areas between shrubs), and then mixed into one soil sample. Soil samples were passed through a 2 mm sieve to remove roots and stones and then returned to the laboratory in an ice box. The soil samples were divided into three parts. One was air-dried to determine the general soil physical and chemical properties, one was stored at −20 °C to determine mineral N (NH_4_^+^−N and NO_3_^−^−N) contents, and one was kept at −20 °C to analyze soil microbial communities.

### 2.4. Determination of Soil Properties

Soil physical and chemical properties were determined according to the description of Zhang et al. [4]. Briefly, soil pH and electrical conductivity (EC) were determined using pH and EC meters at a soil/water ratio of 1:2.5 (*w*/*v*), respectively. SOC was measured with the K_2_Cr_2_O_7_ oxidation method. NH_4_^+^−N and NO_3_^−^−N contents were extracted with 1 M KCl and then measured with the ultraviolet spectroscopy method. AN was measured with the alkali hydrolyzable method. TN was evaluated using the Kjeldahl method. AP was measured by the molybdate/ascorbic acid method. AK was extracted using NH_4_OAc and then measured.

### 2.5. DNA Extraction and Illumina Sequencing

The total DNA was extracted from 0.5 g fresh soil using the DNeasy PowerSoil DNA Isolation Kit (Qiagen, Inc., Venlo, The Netherlands), in accordance with the manufacturer’s protocol. The purity and concentration of the obtained DNA were evaluated using a NanoDrop ND-1000 spectrophotometer (Thermo Fisher Scientific, Waltham, MA, USA). For bacteria, the V3-V4 hypervariable region of the 16S rRNA gene was amplified using the polymerase chain reaction (PCR) primers 338F (5′-ACTCCTACGGGAGGCAGCA-3′)/806R (5′-GGACTACHVGGGTWTCTAAT-3′) [25]. For fungi, the internal transcribed spacer (ITS)1 region was amplified using the PCR primers ITS5 (5′-GGAAGTAAAAGTCGTAACAAGG-3′)/ITS2(5′-GCTGCGTTCTTCATCGATGC-3′) [26]. Sample-specific barcodes were linked to the reverse primers for subsequent amplification. The amplification quality of the PCR products was evaluated by agarose gel electrophoresis. PCR products that met the criteria were then purified and pooled in equal proportions based on DNA concentrations and molecular weight to produce the Illumina^®^ DNA libraries. Paired-end DNA sampling was performed on the Illumina MiSeq System platform (Personal Biotechnology Co., Ltd., Shanghai, China).

For each library, the procedures, including sequence quality control, denoising, splicing, and chimera removal were performed after removing the sequence of unmatched primers in accordance with the DADA2 method [27]. The sequence produced by this method was called amplicon sequence variants (ASVs), which can be used to identify all possible sequences in a sample with great specificity, including those with a single variant. In contrast, the traditional operational taxonomic unit only identifies sequences that are clustered together with a specific degree of similarity [28]. Silva (Release 132) [29] and UNITE (Release 8.0) [30] databases were used to annotate the 16S rRNA and ITS genes, respectively. α-Diversity was assessed with Chao1 index to detect within-sample diversity. The raw data were deposited in the NCBI Sequence Read Archive database under the accession number of PRJNA949150 for bacteria and PRJNA949140 for fungi.

### 2.6. Statistical Analyses

All data analyses were conducted using R software. Abundance analysis was performed for bacterial phyla (92% of total abundance), bacterial classes (62% of total abundance), fungal phyla (100% of total abundance), and fungal classes (97% of total abundance). One-way analysis of variance (ANOVA) and two-way ANOVA were used to investigate the effects of N additions or/and plant community types on microbial abundance, diversity and soil physicochemical properties. The least significant difference (LSD) was used for multiple comparisons. Venn diagrams were used to show the shared and unique ASVs among different plant communities. Microbial community composition at the ASV level was analyzed by principal coordinate analysis combined with permutational multivariate ANOVA (PERMANOVA) with 999 permutations and assessed using the Mantel test. Given the absence of interaction between N addition and plant community type on microbial community composition, Bray–Curtis community dissimilarity was conducted on these two treatments. Redundancy analysis (RDA) combined with hierarchical partitioning was used to analyze the relationship between microbial community composition and soil physicochemical properties [31]. Pearson correlation among microbial diversity, abundance, and soil properties was calculated and visualized via heatmaps, and *p*-values were adjusted using the false discovery rate. PIRCUSt was used to predict the soil bacterial function profiles based on KEGG pathways genes. FUNGuild was used to parse fungal communities by ecological guild. The abundance in the top 20 A was subsequently analyzed. Abundances of these predicted functions were investigated by the effects of N additions within each plant community type by ANOVA. The LSD method was performed to conduct multiple comparisons in the case of significant differences.

## 3. Results

### 3.1. Soil Properties Affected by N Addition

Soil C, N, P, and pH significantly responded to N-addition treatment, plant community type, and their interaction (Table 1; *p* < 0.05). The interaction between N addition and plant community type significantly affected TN, SOC, and AP. Soil TN was the highest in the CK of the *A. sparsifolia* community, whereas in N60 treatment, it was the lowest. The SOC was the highest in the N120-treated *K. caspia* community, while in the N10 treatment, it was the lowest. Soil AP was the highest in the CK of the *A. sparsifolia* community, whereas in the N10-treated mixed community, it was the lowest. N addition, but not plant community type, significantly changed the contents of AN, NH_4_^+^−N, NO_3_^−^−N. Soil AN and NO_3_^−^−N showed a higher level in N60 than in other treatments. The content of NH_4_^+^−N was the highest in CK and N120 treatments. Soil pH was significantly influenced by the independent effect of N addition and plant community type. N addition significantly increased pH, and soil pH was significantly lower in the mixed community than in *A. sparsifolia* and *K. caspia* communities.

### 3.2. Soil Bacterial and Fungal Community Composition

High-throughput sequencing of bacterial and fungal samples generated 55,164–105,226 and 42,633–207,089 sequences, respectively, and eventually produced 5224 (bacteria) and 284 (fungi) ASVs (Appendix A). The contribution of N-addition treatment and plant community type to bacterial abundance and α-diversity (estimated by Chao1) was greater than those of fungi (Figure 2). Actinobacteria, Proteobacteria, and Bacteroidetes dominated the bacterial community (Figure 2a). Compared with CK, N addition significantly changed the abundances of phyla Actinobacteria (31.19% reduction under N60), Bacteroidetes (35.72% increase under N60 and 24.95% reduction under N10), Firmicutes (97.77% increase under N60 and 12.01% reduction under N10), Planctomycetes (29.16% increase under N10), Gemmatimonadetes (26.06% increase under N10 and 19.04% reduction under N120), and Chloroflexi (44.18% increase under N10 and 30.16% reduction under N120) (Figure 2a; *p* < 0.05). Abundances of Alphaproteobacteria and Actinobacteria were significantly reduced by the N60 treatment (Appendix A; *p* < 0.05). The *A. sparsifolia* community showed a significantly lower abundance of phylum Deinococcus-Thermus compared to other community types, while the abundance of Chloroflexia and Deltaproteobacteria was significantly higher in the *A. sparsifolia* community than in other community types (Appendix A; *p* < 0.05). The N60 treatment significantly decreased the abundances of Proteobacteria in the *A. sparsifolia* and mixed communities, but was not significantly affected by N addition in *K. caspia* community (Appendix A; *p* < 0.05). The α-diversity of N30 treatment was the highest in the *A. sparsifolia* and plant mixed communities (Figure 2c; *p* < 0.05). Ascomycota dominated the fungal community (Figure 2b). However, fungal α-diversity showed no significant response to N addition in any community type (Figure 2d; *p* > 0.05).

N addition and plant community type independently and significantly shaped soil bacterial and fungal composition at the ASV level (Figure 3). The PERMANOVA result showed that plant community type (16.29% for bacteria and 14.22% for fungi) contributed more variation to microbial community composition than N addition (11.21% for bacteria and 8.35% for fungi). N addition explained 37.5%, 41.2%, and 35.2% of the variation in bacterial community composition of *A. sparsifolia*, *K. caspia*, and mixed communities, respectively (Appendix A; *p* < 0.05). For fungi, N addition significantly changed the fungal community composition of the *A. sparsifolia* community, but not those of other plant community types (Appendix A). The dissimilarity of the soil bacterial community in the *A. sparsifolia* community was significantly greater than that in other plant communities (Figure 4a; *p* < 0.05). Adversely, the dissimilarity of soil fungal community in the *A. sparsifolia* community was significantly lower than that in other plant communities (Figure 4b; *p* < 0.05). In terms of N-addition treatment, a significantly higher dissimilarity of soil bacterial community was observed under the N60 treatment (Figure 4c; *p* < 0.05). However, no significant difference in the dissimilarity of fungal communities was noted among different N-addition treatments (Figure 4d; *p* > 0.05).

### 3.3. Diversity of Soil Bacterial and Fungal Communities

To identify the factors influencing the formation of soil bacterial and fungal communities across three plant communities subject to varying N-addition treatments, we initially examined the correlation between eight soil properties, dominant phyla and classes, and Chao1. Subsequently, we assessed the connection between soil properties and microbial community composition at the ASV level via the Mantel tests, RDA, and variance partitioning analyses (Figure 5, Figure 6 and Appendix A).

The correlation analysis revealed that the α-diversity of bacteria in the mixed community was significantly positively correlated with AP and pH, while it was significantly negatively correlated with AK (Appendix A; *p* < 0.05), and the α-diversity of fungi in the *K. caspia* community was significantly negatively correlated with NH_4_^+^−N (Appendix A; *p* < 0.05). Soil C, N, and K levels were significantly correlated with the relative abundance of dominant bacterial phyla (i.e., Firmicutes) and classes (i.e., Bacilli) (Appendix A; *p* < 0.05). The significant correlation between the abundance of dominant fungal taxa and soil properties was more observed at the class level (Appendix A; *p* < 0.05).

The Mantel test result showed that pH, AK, and NO_3_^−^−N were powerful factors that significantly shaped the community composition of soil bacteria in mixed communities (Appendix A; Mantel test: >0.308, *p* < 0.05). Soil NH_4_^+^−N and TN explained more variation than any of the other factors in the fungal community composition of *A. sparsifolia* and *K. caspia* communities, respectively (Mantel test: >0.254, *p* < 0.05).

RDA and variance partitioning analyses suggested that soil NO_3_^−^−N, TN, and pH had the highest individual importance in explaining the variation in the community composition of bacteria in *A. sparsifolia*, *K. caspia*, and their mixed communities (Figure 5). Soil NH_4_^+^−N had the highest individual effect on the fungal community composition of the *A. sparsifolia* community (Figure 6c). Soil pH was the strongest determinant of fungal community composition in *K. caspia* and their mixed communities (Figure 6e,f).

### 3.4. Function Prediction of Soil Bacterial and Fungal Communities

According to the PICRUSt result for the bacterial community, there were 24 predicted KEGG functional genes whose abundances significantly differed among five N-addition treatments within each plant community (Figure 7, *p* < 0.05). Of those, the dominant potential function in three plant community types was “metabolism”, accounting for 23.7–25.1%. N addition significantly influenced the abundance of amino acid metabolism and xenobiotics biodegradation and metabolism in three plant community types. In contrast, only three fungal guild abundances significantly differed among N-addition treatments (Figure 8, *p* < 0.05). Specifically, N addition significantly decreased the abundance of “endophyte-Plant Pathogen-Undefined Saprotroph” in the *A. sparsifolia* community. However, abundances of “animal endosymbiont-animal pathogen-plant pathogen-undefined saprotroph” and “plant saprotroph-wood saprotroph” were increased by N60 and N10 treatments in the *K. caspia* community, respectively.

## 4. Discussion

Numerous studies have investigated the response of soil microbial communities to N deposition or plant community types. However, only a limited number of studies have considered these factors together in desert ecosystems. Our study explored the potential effect of N inputs and plant community type on soil bacterial and fungal community structure in the desert steppe at the southern edge of the Taklimakan Desert. Increased and then decreased bacterial α-diversity and community dissimilarity were observed with increasing N additions, whereas the fungal community was more insensitive, partially supporting our first hypothesis. The independent effects of N deposition and plant community type significantly altered soil microbial community composition, which does not support our second hypothesis. This finding indicates that increased N deposition alters soil microbial community structure, and this effect is independent of plant community types in this region. Soil pH and the form and amount of N are important factors regulating soil microbial community.

### 4.1. Effects of N Addition on Soil Microbial Community Structure

Given the close relevance of soil microorganisms to biogeochemical cycles, it is important to understand the mechanisms by which soil microbial abundance, diversity and composition respond to environmental changes [20]. Our results revealed a unimodal and domain-specific diversity pattern for the soil microbial community with elevated N deposition in this region. Previous studies have shown that soil bacterial or/and fungal diversity in desert ecosystems is insensitive to elevated N deposition [16,17,32]. In this study, although the community composition of soil bacteria and fungi was significantly affected by N deposition, their diversity and community dissimilarity responded differently. Bacterial diversity and community dissimilarity first significantly increased and then decreased with increasing N deposition rate, whereas fungal diversity was stable under the same environmental conditions (Figure 2c,d and Figure 4c,d). These results indicate that there is a threshold in the response of bacterial diversity and community dissimilarity to N addition, and that fungi in desert soils are more tolerant to N deposition than bacteria. The nonlinear responses of bacterial diversity to N addition can be attributed to the availability of soil resources and competition for resources between bacterial species. Low levels of N addition can alleviate N limitation, promoting bacterial growth and leading to an increase in diversity [33]. As N availability increases, certain bacterial taxa may have competitive advantages over others, leading to dominance and a subsequent decrease in diversity [34,35]. The initial stimulation and subsequent competitive exclusion mechanism may result in a nonlinear response, with diversity peaking at intermediate N levels (Figure 2c). The different responses of bacterial and fungal communities to N addition may reflect their various adaptation strategies in desert habitats. Fungi have better access to soil nutrients through their mycelial network [36], whereas these mycelia may be damaged by an arid environment [37], reducing their sensitivity to resource changes. Alternatively, unchanged fungal diversity and community dissimilarity suggest a frequent association of a core soil fungal microbiome with different N-deposition scenarios, which may play a particularly important role in N-cycle-associated soil functions. The strong fungal community response to N enrichment may occur in long-term experimental treatments.

N addition significantly affected the composition of bacterial and fungal communities (Figure 3), which is consistent with previous studies conducted in other desert steppe ecosystems [3,16,38]. There are several possible explanations for this change before and after N addition. Firstly, previous studies conducted in semi-arid steppes [14] and on a global scale [39] have documented that decreased pH induced by N fertilization plays a vital role in the soil microbial community [38]. Although our Mantel test and RDA also suggested that pH induced by N addition significantly influenced microbial community composition, pH in our case showed an increasing trend after N additions (Table 1 and Appendix A; Figure 5 and Figure 6). The up-regulation of pH after N addition may be related to the combined effects of soil type, initial pH, buffering capacity, and the specific form and amount of N added [40,41,42]. For example, urea can accumulate large amounts of ammonium ions in the soil, which combine with water to form ammonium hydroxide, leading to soil alkalinization [43,44]. Secondly, N additions can increase NO_3_^−^−N content by increasing potential nitrification rates and improving nitrate reductase [14], which can regulate the soil microbial community induced by plant growth under increased soil nutrients (e.g., AN) [4]. Our experiment revealed that the TN level and N form (i.e., NH_4_^+^−N and NO_3_^−^−N) were the main regulators of soil microbial community composition (Figure 5 and Figure 6; Table 1).

The response of copiotrophic bacteria to N addition is different from that of oligotrophic taxa. Copiotrophic bacteria can easily consume liable C with a highly additional N input to support their high intrinsic growth rates, whereas oligotrophic bacteria primarily consume less absorbable C using less nutrients and therefore grow at a slower rate [45]. Our results partially support this hypothesis. In this study, the level of Firmicutes, which belongs to a copiotrophic group, was improved by N addition (N60), whereas those of Actinobacteria and Proteobacteria, which also belong to a copiotrophic group, were significantly decreased under N-addition treatments (Appendix A). These results suggest that changes in the abundance of copiotrophic and oligotrophic bacteria induced by N addition were inconsistent within each functional group [3]. This finding is supported by two possible explanations: firstly, multiple variables (i.e., soil moisture and vegetation cover) that influence bacterial growth and life histories may lead to the inconsistency of copio-oligotrophic associates at the phylum or class level [46]. Firmicutes are better adapted to extreme environments because they can produce endospores that help tolerate dehydrating conditions [47]. Their abundance was significantly improved by a suitable external N supply (N60 in this study), which likely reduced the abundance of other copiotrophic taxa. Secondly, even within the same phylum, differences in life history and growth can lead to various ecological functions and niches. The phylum Proteobacteria contain numerous functionally diverse taxa, such as Alphaproteobacteria, Gammaproteobacteria, and Deltaproteobacteria. In our study, only Alphaproteobacteria significantly responded to N addition (Appendix A). As a shortcoming of this study, given the short experimental processing time, an extended observation period is needed in the future to clarify the direct and indirect effects of global changes on soil microbial communities in desert ecosystems.

### 4.2. Effects of Plant Community Types on Soil Microbial Community Structure

Different plant community types may affect soil microbial community structure, thereby affecting their response to nutrient additions [36,48]. However, the small relationship between plant community types and soil microbial diversity has been documented in several studies [45,49]. In our research, plant community types significantly altered soil microbial community composition, but did not affect their diversity (Figure 2c,d and Figure 3b). This finding may be attributed to differences in the quantity and quality of plant-derived resources (e.g., litters and root exudates) in the soil and the resulting differences in soil physical and chemical properties [7]. However, these changes in edaphic factors were insufficient to alter microbial diversity.

Soil pH can regulate the availability of nutrients in soil and is closely related to biogeochemical cycles [50]. In the present study, a lower soil pH was observed in the mixed community than the pure communities (Table 1), which largely contributed to the variation in the composition of soil microbial community in the investigated plant community type (Table 1; Figure 5c,f and Figure 6c,f). This result may be due to three potential mechanisms. Firstly, the differences in nutrient uptake and storage of different plants may affect soil acidity. Different plant species have varying nutrient uptake patterns and requirements [51]. In mixed plant communities, certain species may have higher nutrient demands or preferentially take up specific nutrients from the soil. As plants uptake nutrients, this can result in the depletion of certain ions, such as calcium (Ca) and magnesium (Mg), which are often linked to soil pH-buffering capacity. The extraction of these ions from the soil leads to a reduction in the soil’s pH level. [52]. Secondly, plants in mixed communities contribute different types and quantities of litter to the soil. During the decomposition process of litter, organic acids such as acetic, oxalic, and citric acid are commonly released into the soil, leading to soil acidification and a decrease in pH [53,54,55]. *A. sparsifolia* or *K. caspia* may have litters with higher concentrations of these organic acids, leading to a more pronounced effect on soil pH when present in a mixed plant community. Thirdly, in mixed plant communities, intense P deficiency induced by strong competition for N among plant–microbe and microbe–microbe communities can influence soil pH through microbial-mediated processes [56]. When available phosphorus (AP) in soil is insufficient, phosphate solubilizing can dissolve inorganic phosphorus compounds by secreting organic acids (e.g., citramalic acid and salicylic acid) and enzymes, converting them into dissolved phosphorus that can be absorbed and utilized by plants, which induce soil acidity [57]. Our RDA results indicate that AP is the second most important factor affecting the soil fungal community (Figure 5c,f). Such findings can be expected given the critical role of fungi in desert soil P cycle [58]. Furthermore, soil bacteria diversity showed a significantly positive correlation with pH and AP in the mixed community, although the bacterial diversity was slightly affected by plant community types (Figure 2d and Appendix A).

Plant community characteristics (e.g., cover and plant identity) can influence the abundance of certain bacterial groups. Class Chloroflexia includes phototrophic bacteria, and their abundance is the best predictor of soil N content [59]. Our result revealed a significantly higher Chloroflexia abundance in the *A. sparsifolia* community than in other community types (Appendix A). The difference can be attributed to the higher transmittance under the shrub due to the smaller leaf area and thinner canopy [60], as well as the biological N fixation by legume *A. sparsifolia* [61].

Except for the fungal community in the *K. caspia* monoculture community, the edaphic factors influencing soil bacterial and fungal community composition differed in various plant community types (Table 1; Figure 5 and Figure 6). These results suggest that soil microbial communities and the biogeochemical cycles they mediate (e.g., nitration and organic matter decomposition) may be modified by plant community types in a desert steppe. Consequently, perennial desert plant community types can indirectly shape the soil microbial community via soil pH and nutrient availability. However, further investigations involved with higher levels of diversity and exploration of litter properties are necessarily separating their direct and indirect effects on soil microbial community.

In our research area, the influence of sandstorms on topsoil texture and rainfall events on ephemeral plants were uncertainties impairing the experimental results (e.g., the potential interaction between N additions and plant community types). Our subsequent experiments will systematically explore the effects of N deposition and plant community types on soil microbial communities under more refined and controlled conditions. Although several flaws should be optimized in our future research, our present study revealed that the response of soil microbial communities to N deposition is domain-specific and independent of desert plant community diversity. The bacterial community has a critical threshold under different N scenarios in an arid desert.

### 4.3. Effects of N Addition and Plant Community Types on the Function of Soil Microbial Community

The enrichment of N in soil changed the structure of the soil microbial community and inevitably caused a variation in soil microbial community function. Our prediction showed that N addition substantially altered bacterial function profiles within each plant community type (Figure 7). However, only three fungal functional guilds were markedly influenced by N addition (Figure 8). These findings were consistent with the variation in their taxonomic abundance and community composition (Figure 2 and Figure 4).

Amino acids have significant roles not only as integral constituents of proteins but also in numerous essential physiological processes. Furthermore, they act as precursors for various secondary metabolites and possess crucial functions such as plant signaling, stress defense, and interaction with other organisms [62]. In this study, the N-addition rate of 60 kg N ha^−1^ year^−1^ (N60) may damage the above processes of microorganisms and be independent of the type of plant community in which they are located (Figure 7). The pathways involved in nucleotide metabolism play a significant role in aiding soil bacteria to adapt to environmental stress conditions. The upregulation of this gene abundance under N60 and N120 conditions suggests that future N enrichment is expected to harm the adaptability of bacteria to extreme habitats [63]. Saprotrophic fungi are crucial contributors to the process of decomposition because they possess the ability to break down the lignocellulose matrix found in the litter. In our study, the N addition rate of 10 kg N ha^−1^ year^−1^ (N10) significantly upregulated the abundance of genes associated with “plant saprotroph–wood saprotroph”, which was consistent with the previous study [15].

Similar to the findings on the taxonomic composition, the fungal functionality exhibits stronger resistance to N addition than bacteria. This trend is also confirmed by the previous research [64]. To investigate the soil microbial function profiles in the desert steppe under N addition, further research is required that employs metagenomic, transcriptomic, proteomic, or metabolomic data.

## 5. Conclusions

Our study revealed that short-term N addition and different types of plant communities independently and significantly shaped the composition of soil bacterial and fungal communities via changes in soil pH and nutrient content. With the increase in the N-deposition rate, bacterial diversity and community dissimilarity showed a unimodal pattern. By contrast, N deposition did not affect fungal dominant phyla and classes, diversity, and community dissimilarity. This insensitivity may contribute to the buffering of desert steppe ecosystems against environmental changes. Plant community types did not exert statistical influence on soil microbial diversity, whereas soil pH and TN and AN content significantly affected soil microbial community composition. Fungal functionality demonstrates a greater resistance to N addition than bacteria, similar to the patterns observed in taxonomic composition. Our findings suggest that N deposition and plant community types regulate soil microbial communities through noninteractive pathways in this desert steppe ecosystem. This understanding will enhance our ability to predict the response of plant and microbial communities to the future state of N deposition and global climate change in desert steppe ecosystems.

## Figures and Tables

**Figure 1 microorganisms-11-02471-f001:**
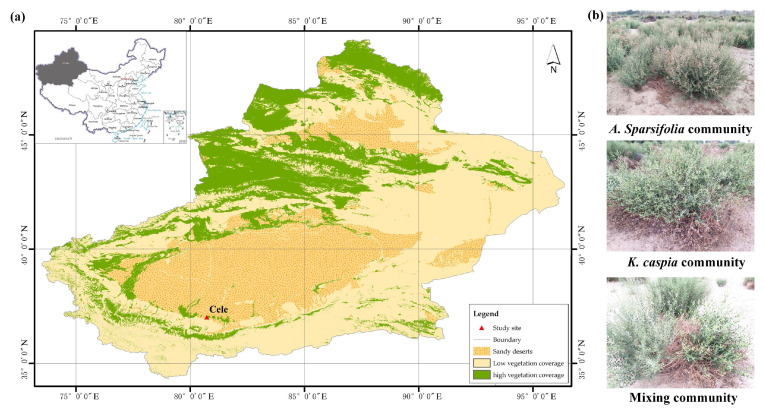
(**a**) The location of the study site. (**b**) Photographs of the three communities.

**Figure 2 microorganisms-11-02471-f002:**
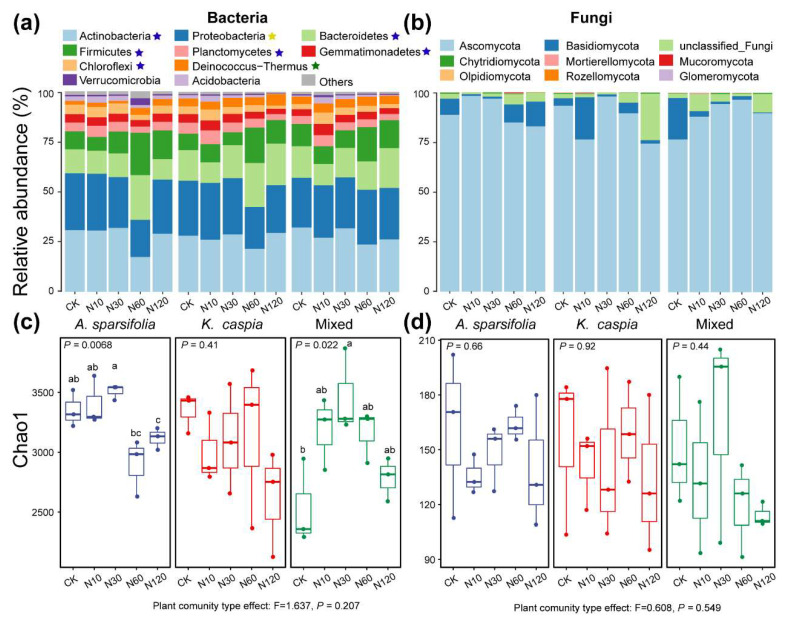
Community composition of bacteria (**a**) and fungi (**b**) at the phylum level (top 10). The blue, green, and yellow pentacle in the legend indicates that the variation in abundance was significant in response to N-addition treatment, plant community, and their interaction (LSD, *p* < 0.05). The α-diversity (estimated by Chao1) of bacteria (**c**) and fungi (**d**). Different lowercase letters indicate a significant difference among different N-addition treatments in each plant community (LSD, *p* < 0.05).

**Figure 3 microorganisms-11-02471-f003:**
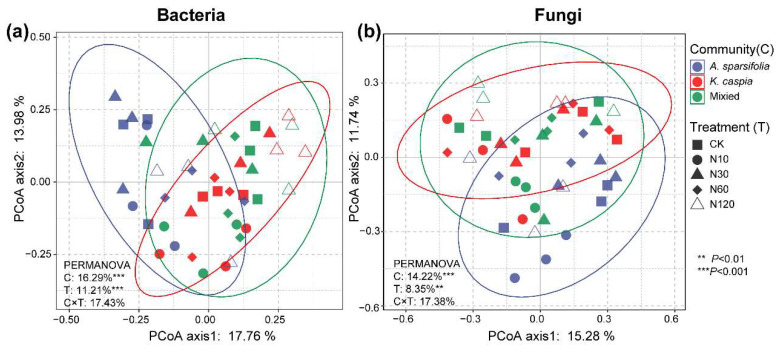
Principal coordinate analysis (PCoA) of the bacterial (**a**) and fungal (**b**) communities based on the Bray–Curtis distance of ASV. The ellipse represents the 95% confidence interval. Permutational analysis of variance (PERMANOVA) with 999 permutations shows the effects of N-addition treatment (T) and plant community type (C) on microbial community composition based on the Bray–Curtis distance of ASVs.

**Figure 4 microorganisms-11-02471-f004:**
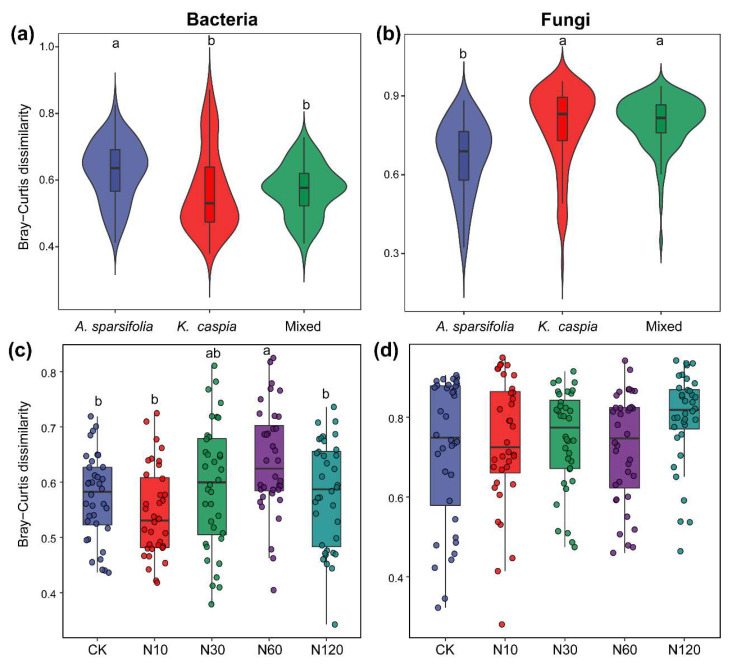
The dissimilarity distances of bacterial (**a**,**c**) and fungal (**b**,**d**) communities under different plant community types and N-addition treatments. Different lowercase letters indicate a significant difference among plant communities and N-addition treatments (Kruskal–Wallis test or LSD, *p* < 0.05).

**Figure 5 microorganisms-11-02471-f005:**
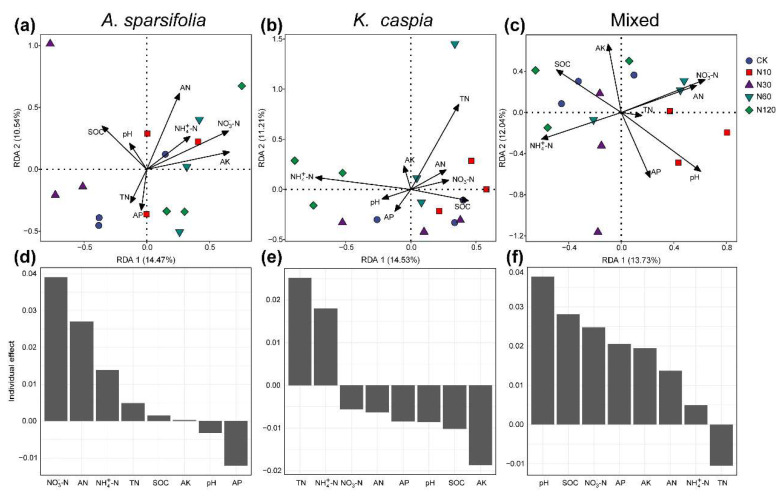
Redundancy analysis (RDA) of the bacterial community in *A. sparsifolia*- (**a**) and *K. caspia*-communities (**b**) and their mixed community (**c**). The relative importance of individual soil physicochemical parameters in predicting the bacterial community composition in each plant community (**d**–**f**). SOC, soil organic carbon; AP, available phosphorous; AK, available potassium; TN, total nitrogen; NO_3_^−^−N, nitrate; NH_4_^+^−N, ammonium; AN, available nitrogen.

**Figure 6 microorganisms-11-02471-f006:**
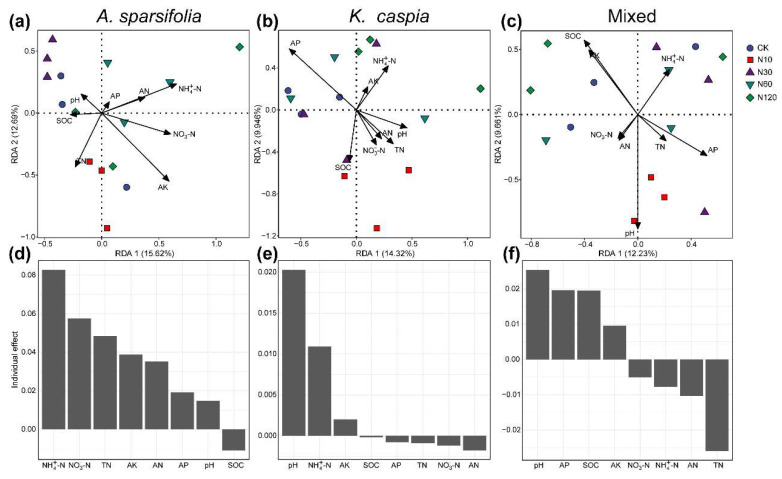
Redundancy analysis (RDA) of fungal community in *A. sparsifolia*- (**a**) and *K. caspia*-communities (**b**) and their mixed community (**c**). The relative importance of individual soil physicochemical parameters in predicting the fungal community composition in each plant community (**d**–**f**). SOC, soil organic carbon; AP, available phosphorous; AK, available potassium; TN, total nitrogen; NO_3_^−^−N, nitrate; NH_4_^+^−N, ammonium; AN, available nitrogen.

**Figure 7 microorganisms-11-02471-f007:**
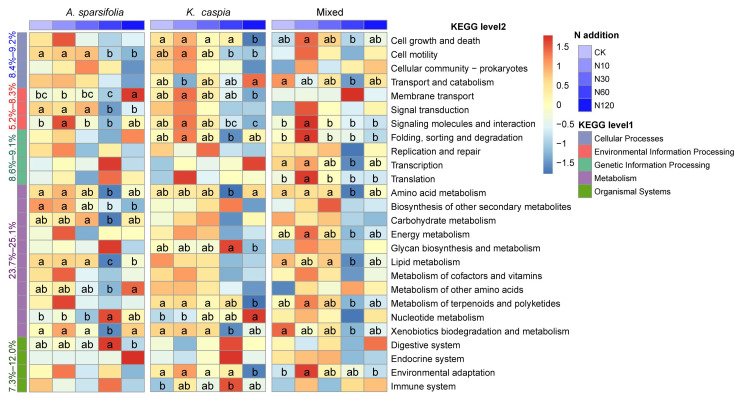
Variation in abundances of soil bacterial function profiles in three plant community types under different N-addition treatments analyzed by PICRUSt. Different lowercase letters indicate a significant difference among five N-addition treatments at 0.05 level.

**Figure 8 microorganisms-11-02471-f008:**
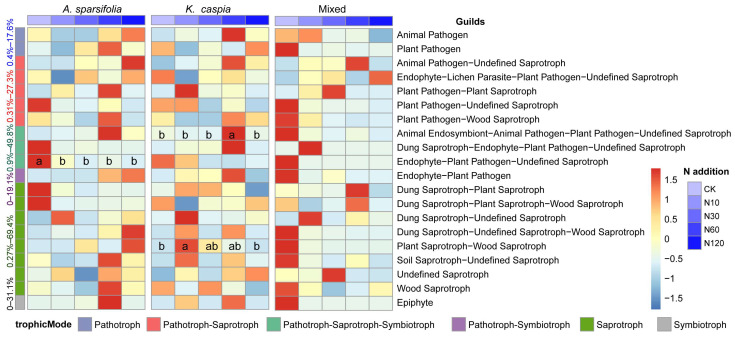
Variation in abundances of soil fungal functional guilds in three plant community types under different N-addition treatments. Different lowercase letters indicate a significant difference among five N-addition treatments at 0.05 level.

**Table 1 microorganisms-11-02471-t001:** Effects of nitrogen-addition treatment, plant community types, and their interaction on soil properties.

N-Addition Treatment × Plant Community Type					
		TN	SOC	AP	AK
*A. sparsifolia*	CK	0.290 ± 0.040 a	1.71 ± 0.10 b	6.11 ± 0.71 a	201.13 ± 8.37 a
	N10	0.186 ± 0.041 def	1.65 ± 0.35 b	3.81 ± 0.39 ab	244.15 ± 16.95 a
	N30	0.160 ± 0.033 ef	1.88 ± 0.35 ab	4.38 ± 1.35 ab	178.83 ± 21.65 a
	N60	0.150 ± 0.046 f	1.38 ± 0.24 bc	4.45 ± 0.82 ab	228.80 ± 33.56 a
	N120	0.185 ± 0.033 def	1.40 ± 0.27 bc	4.24 ± 0.90 ab	205.40 ± 17.45 a
*K. caspia*	CK	0.273 ± 0.048 ab	1.34 ± 0.02 bc	5.41 ± 1.59 ab	226.03 ± 18.94 a
	N10	0.264 ± 0.017 abc	1.81 ± 0.12 ab	2.99 ± 0.41 b	210.06 ± 18.60 a
	N30	0.197 ± 0.036 cdef	1.77 ± 0.22 ab	4.06 ± 0.49 ab	221.66 ± 24.23 a
	N60	0.238 ± 0.014 abcd	1.48 ± 0.14 bc	5.01 ± 1.39 ab	246.43 ± 34.71 a
	N120	0.229 ± 0.035 abcde	1.46 ± 0.38 bc	3.77 ± 1.03 ab	234.03 ± 41.39 a
Mixing	CK	0.185 ± 0.029 def	1.44 ± 0.23 bc	3.69 ± 0.60 ab	243.66 ± 35.22 a
	N10	0.238 ± 0.030 abcd	1.00 ± 0.16 c	4.77 ± 0.19 ab	216.40 ± 40.17 a
	N30	0.182 ± 0.022 def	1.34 ± 0.20 bc	5.34 ± 1.56 ab	206.76 ± 27.73 a
	N60	0.232 ± 0.044 abcde	1.61 ± 0.33 b	5.28 ± 0.75 ab	231.56 ± 11.71 a
	N120	0.215 ± 0.024 bcdef	2.29 ± 0.31 a	4.67 ± 0.63 ab	240.13 ± 7.82 a
**N-addition treatment**					
	AN	NH_4_^+^−N	NO_3_^−^−N	pH	
CK	28.9 ± 8.5 b	3.8 ± 0.8 a	19.7 ± 7.9 b	7.11 ± 0.06 c	
N10	37.6 ± 13.9 ab	3.1 ± 0.2 b	30.1 ± 11.8 ab	7.51 ± 0.13 a	
N30	31.0 ± 9.70 b	3.6 ± 0.8 ab	18.6 ± 9.9 b	7.49 ± 0.20 a	
N60	46.9 ± 16.8 a	3.5 ± 0.4 ab	34.0 ± 16.5 a	7.28 ± 0.09 b	
N120	25.4 ± 15.0 b	4.2 ± 1.0 a	18.8 ± 13.9 b	7.38 ± 0.21 ab	
**Plant community type**					
	pH				
*A. sparsifolia*	7.37 ± 0.19 b				
*K. caspia*	7.41 ± 0.20 a				
Mixing	7.29 ± 0.21 c				

Note: SOC, soil organic carbon, g kg^−1^; AP, available phosphorus, mg kg^−1^; AK, available potassium, mg kg^−1^; TN, total nitrogen, g kg^−1^; NO_3_^−^−N, nitrate, mg kg^−1^; NH_4_^+^−N, ammonium, mg kg^−1^; AN, available nitrogen, mg kg^−1^. Different lowercase letters in the same column indicate significant differences between groups (*p* < 0.05).

## Data Availability

The data presented in this study are available on request from the corresponding author. The data are not available to the public due to the data management requirements of the Chinese Academy of Sciences.

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
