# Peer review of "Different Responses of Soil Bacterial and Fungal Communities in Three Typical Vegetations following Nitrogen Deposition in an Arid Desert"

_microorganisms, 2023, doi:10.3390/microorganisms11102471_

Round 1
Reviewer 1 Report
The described study is of international relevance, because fully cover all important tasks regarding the response of soil microbial communities, which play a crucial role in nutrient cycling, to N inputs and plant community types. The obtained results indicate that N deposition and plant community types exerted an independent and significant influence on soil microbial community. The article is well written for the most part and should be of interest to readers. The result and discussion was in-depth. Statistical tools were used. The article is devoted to a very relevant topic and deserves publication in the journal. Therefore, can be considered for publication after slight revision.
- The thinking behind the research in the manuscript is clear. However, it is needs to clarify the research's innovation and process.
-Part of the introdution section is not relevant to the research objectives. I suggest reduce or delete some parts.
-I suggest to include some experimental photos.
I wish those changes will contribute to improve your paper.
Author Response
Response to Reviewer 1 Comments
Dear Reviewer,
Thank you for your comments concerning the manuscript entitled "Different responses of soil bacterial and fungal communities in three typical vegetation following nitrogen deposition in an arid desert" (ID: 2608444). These comments are all valuable and very helpful for revising and improving our paper, as well as the important guiding significance to our researchers. We have studied the comments carefully and made a correction that we hope meets with approval. We revised the manuscript using the revision mode and prepared a clean version. The main corrections in the paper and the responses to your comments are as follows:
Point 1: The thinking behind the research in the manuscript is clear. However, it is needs to clarify the research's innovation and process.
Response 1: Thanks for your friendly suggestion. We have clarified the manuscript's innovation and process and included this information in the Introduction as following:
Lines 73-76: Currently, there is not a clear understanding of how soil microorganisms respond to N addition in the typical desert plant communities at the southern edge of the Taklamakan Desert in extremely arid areas. In this area, the vegetation is mainly composed of deep-rooted plants such as Alhagi sparsifolia Shap. and Karelinia caspia (Pall.) Less.
In addition, our findings suggest that N deposition and plant community types regulate soil microbial communities through noninteractive pathways in the desert steppe ecosystem. This understanding will enhance our ability to predict the response of plant and microbial communities to the future state of N deposition and global climate change in desert steppe ecosystems (lines 515-519).
Point 2: Part of the introdution section is not relevant to the research objectives. I suggest reduce or delete some parts.
Response 2: We have deleted the parts (lines 73-86 in the previous manuscript) that is not relevant to the research objectives.
Point 3: I suggest to include some experimental photos.
Response 3: We have included the experimental photos in line 108 as follows:
Figure 1 (a)The location of the study site. (b) Photographs of the three communities.

Reviewer 2 Report
Specific comments:
- Please correct the literature citation as required in the journal.
- Please add the country (China), (L. 111).
- Why were soil samples taken only once – in 2020?
- Please provide the soil characteristics before the experiment (Chapter 2.4).
- The results of individual parameters should be presented in a unified manner following the Materials and methods. In Table 1, in the case of AN, NH4+ −N, NO3- –N, pH, one of the factors was not taken into account - A. sparsifolia monoculture, K. caspia monoculture, and their mixed communities. Results and Discussion should be changed accordingly.
- There is no AK in Table 1 (L. 225). If the authors included potassium in their research (Figs 4 and 5, Table S1), please provide this information in the materials and methods and Table 1.
- Please correct editing errors, e.g. Secondly, Plants (…) (L. 456).
Minor editing of English language is required.
Author Response
Response to Reviewer 2 Comments
Dear Reviewer,
Thank you for your comments concerning the manuscript entitled “Different responses of soil bacterial and fungal communities in three typical vegetation following nitrogen deposition in an arid desert” (ID: 2608444). These comments are all valuable and very helpful for revising and improving our paper, as well as the important guiding significance to our researchers. We have studied the comments carefully and made a correction that we hope meets with approval. We revised the manuscript using the revision mode and prepared a clean version. The main corrections in the paper and the responses to your comments are as follows:
Point 1: Please correct the literature citation as required in the journal.
Response 1: Thank you for the kind reminder. We have modified the style of the journal literature.
Point 2: Please add the country (China), (L. 111).
Response 2: We have added the country (China) where the study area is located in line 97 and annotated it in Figure 1a as following:
Lines 96-98: This study was carried out in a desert steppe (80°43′E, 37°00′N, 1363 m a.s.l.) at the southern edge of Taklimakan Desert, China, the second-largest drifting desert in the world (Figure 1a).
Figure 1. (a)The location of the study site. (b) Photographs of the three communities.
Point 3: Why were soil samples taken only once – in 2020?
Response 3: The experiment was conducted in 2020 to explore the effects of nitrogen adding to soil microbes in the short term (two years), so soil samples are only taken from 2020. At the same time, we are also exploring the impact of long-term nitrogen addition on the soil microbial community, which are sampled each year, and the concrete results will be published in the future.
Point 4: Please provide the soil characteristics before the experiment (Chapter 2.4).
Response 4: We have provided the soil characteristics before the treatments in chapter 2.4 as following:
Lines 124-128: Before experimental treatment, the properties of soil were as follows: soil organic carbon (SOC), 1.50 g kg−1; total nitrogen (TN), 0.25 g kg−1; nitrate (NO− 3−N), 19.74 mg kg−1; ammonium (NH+ 4−N), 3.84 mg kg−1; available nitrogen (AN), 28.87 mg kg−1; available phosphorus (AP), 5.07 mg kg−1; available potassium (AK), 223.61 mg kg−1; pH, 7.11.
Point 5: The results of individual parameters should be presented in a unified manner following the Materials and methods. In Table 1, in the case of AN, NH4+ −N, NO3- –N, pH, one of the factors was not taken into account - A. sparsifolia monoculture, K. caspia monoculture, and their mixed communities. Results and Discussion should be changed accordingly.
Response 5: Thank you for your suggestion. We are sorry for our unclear expression. In fact, the soil indicators in Table 1 were divided into three categories: the first category was affected by the interaction between N addition and plant community type; the second category was only affected by N addition; and the third category was only affected by plant community type. This classification provided a clear picture of the effects of N additions and plant community types on soil properties.
Point 6: There is no AK in Table 1 (L. 225). If the authors included potassium in their research (Figs 4 and 5, Table S1), please provide this information in the materials and methods and Table 1.
Response 6: We have added the determination method of AK in Materials (lines 146-147) and Methods and the data in Table 1.
Lines: 142-143AK was extracted using NH4OAc and measured.
Table 1. Effects of nitrogen addition treatment, plant community types, and their interaction on soil properties.
|
N addition treatment × plant community type |
|||||
|
TN |
SOC |
AP |
AK |
||
|
A. sparsifolia |
CK |
0.290±0.040a |
1.71±0.10b |
6.11±0.71a |
201.13±8.37a |
|
N10 |
0.186±0.041def |
1.65±0.35b |
3.81±0.39ab |
244.15±16.95a |
|
|
N30 |
0.160±0.033ef |
1.88±0.35ab |
4.38±1.35ab |
178.83±21.65a |
|
|
N60 |
0.150±0.046f |
1.38±0.24bc |
4.45±0.82ab |
228.80±33.56a |
|
|
N120 |
0.185±0.033def |
1.40±0.27bc |
4.24±0.90ab |
205.40±17.45a |
|
|
K. caspia |
CK |
0.273±0.048ab |
1.34±0.02bc |
5.41±1.59ab |
226.03±18.94a |
|
N10 |
0.264±0.017abc |
1.81±0.12ab |
2.99±0.41b |
210.06±18.60a |
|
|
N30 |
0.197±0.036cdef |
1.77±0.22ab |
4.06±0.49ab |
221.66±24.23a |
|
|
N60 |
0.238±0.014abcd |
1.48±0.14bc |
5.01±1.39ab |
246.43±34.71a |
|
|
N120 |
0.229±0.035abcde |
1.46±0.38bc |
3.77±1.03ab |
234.03±41.39a |
|
|
Mixing |
CK |
0.185±0.029def |
1.44±0.23bc |
3.69±0.60ab |
243.66±35.22a |
|
N10 |
0.238±0.030abcd |
1.00±0.16c |
4.77±0.19ab |
216.40±40.17a |
|
|
N30 |
0.182±0.022def |
1.34±0.20bc |
5.34±1.56ab |
206.76±27.73a |
|
|
N60 |
0.232±0.044abcde |
1.61±0.33b |
5.28±0.75ab |
231.56±11.71a |
|
|
N120 |
0.215±0.024bcdef |
2.29±0.31a |
4.67±0.63ab |
240.13±7.82a |
|
|
N addition treatment |
|
||||
|
AN |
NH+ 4−N |
NO− 3−N |
pH |
|
|
|
CK |
28.9±8.5b |
3.8±0.8a |
19.7±7.9b |
7.11±0.06c |
|
|
N10 |
37.6±13.9ab |
3.1±0.2b |
30.1±11.8ab |
7.51±0.13a |
|
|
N30 |
31.0±9.70b |
3.6±0.8ab |
18.6±9.9b |
7.49±0.20a |
|
|
N60 |
46.9±16.8a |
3.5±0.4ab |
34.0±16.5a |
7.28±0.09b |
|
|
N120 |
25.4±15.0b |
4.2±1.0a |
18.8±13.9b |
7.38±0.21ab |
|
|
Plant community type |
|
||||
|
pH |
|
||||
|
A. sparsifolia |
7.37±0.19b |
|
|
||
|
K. caspia |
7.41±0.20a |
|
|||
|
Mixing |
7.29±0.21c |
|
|
||
Note: SOC, soil organic carbon, g kg−1; AP, available phosphorus, mg kg−1; AK, available potassium, mg kg−1; TN, total nitrogen, g kg−1; NO− 3−N, nitrate, mg kg−1; NH+ 4−N, ammonium, mg kg−1; AN, available nitrogen, mg kg−1. Different lowercase letters in the same column indicate significant differences between groups (P<0.05).
Point 7: Please correct editing errors, e.g. Secondly, Plants (…) (L. 456).
Response 7: We have corrected this editing error as following:
Lines 438-439: Secondly, plants in mixed communities contribute different types and quantities of litter to the soil.
Point 8 Minor editing of English language is required.
Response 8: We have polished the language using Shinewriye.com. We have scrutinized the manuscript, and made according revisions including some typos, grammatical errors and long sentences in the revised manuscript.
